# Multi-Method Diagnosis of Blood Microscopic Sample for Early Detection of Acute Lymphoblastic Leukemia Based on Deep Learning and Hybrid Techniques

**DOI:** 10.3390/s22041629

**Published:** 2022-02-19

**Authors:** Ibrahim Abunadi, Ebrahim Mohammed Senan

**Affiliations:** 1Department of Information Systems, College of Computer and Information Sciences, Prince Sultan University, Riyadh 11586, Saudi Arabia; 2Department of Computer Science & Information Technology, Dr. Babasaheb Ambedkar Marathwada University, Aurangabad 431004, India

**Keywords:** acute lymphoblastic leukemia, machine learning, convolutional neural network, hybrid method, local binary pattern, gray level co-occurrence matrix, fuzzy color histogram

## Abstract

Leukemia is one of the most dangerous types of malignancies affecting the bone marrow or blood in all age groups, both in children and adults. The most dangerous and deadly type of leukemia is acute lymphoblastic leukemia (ALL). It is diagnosed by hematologists and experts in blood and bone marrow samples using a high-quality microscope with a magnifying lens. Manual diagnosis, however, is considered slow and is limited by the differing opinions of experts and other factors. Thus, this work aimed to develop diagnostic systems for two Acute Lymphoblastic Leukemia Image Databases (ALL_IDB1 and ALL_IDB2) for the early detection of leukemia. All images were optimized before being introduced to the systems by two overlapping filters: the average and Laplacian filters. This study consists of three proposed systems as follows: the first consists of the artificial neural network (ANN), feed forward neural network (FFNN), and support vector machine (SVM), all of which are based on hybrid features extracted using Local Binary Pattern (LBP), Gray Level Co-occurrence Matrix (GLCM) and Fuzzy Color Histogram (FCH) methods. Both ANN and FFNN reached an accuracy of 100%, while SVM reached an accuracy of 98.11%. The second proposed system consists of the convolutional neural network (CNN) models: AlexNet, GoogleNet, and ResNet-18, based on the transfer learning method, in which deep feature maps were extracted and classified with high accuracy. All the models obtained promising results for the early detection of leukemia in both datasets, with an accuracy of 100% for the AlexNet, GoogleNet, and ResNet-18 models. The third proposed system consists of hybrid CNN–SVM technologies, consisting of two blocks: CNN models for extracting feature maps and the SVM algorithm for classifying feature maps. All the hybrid systems achieved promising results, with AlexNet + SVM achieving 100% accuracy, Goog-LeNet + SVM achieving 98.1% accuracy, and ResNet-18 + SVM achieving 100% accuracy.

## 1. Introduction

Blood is one of the significant elements of the human body, consisting of 55% plasma and 45% red blood cells (RBCs) [1]. There are also white blood cells (WBCs) and platelets, comprising less than 1% of the blood. There are three main blood components according to the shape, color, size, composition, and texture of the blood: RBC [2], WBC [3], and platelets [4]. RBC is the most important blood sample, and hemoglobin is one of its basic components, giving blood its red color and transporting oxygen to all parts of the body. When hemoglobin levels are reduced, oxygen decreases, causing fatigue and weakness. The RBC rate ranges from 4,000,000 to 6,000,000 per microliter of blood, representing 40–45% of the total blood volume [5]. WBCs are the cells that defend against germs and give us immunity and resistance; they range from 4500 to 11,000 per microliter of blood [6]. The platelets range from 150,000 to 450,000 per microliter of blood and are responsible for blood clotting [7]. Thus, an increase or decrease in any of the basic blood components will cause problems to a person’s health, such as leukemia, thalassemia, and anemia. A high WBC volume leads to poor body immunity because it covers both the RBCs and platelets. Medical practitioners classify it into four types according to its development, speed, and impact: acute lymphoblastic leukemia (ALL), chronic lymphocytic leukemia, acute myeloblastic leukemia, and chronic myeloblastic leukemia. Of these, ALL is the most common, representing 70% of all leukemia cases, and the most fatal. Additionally, environmental and genetic factors have an active role in the development of the disease. The cause of ALL is the excessive and uncontrollable proliferation of lymphocytes in the bone marrow [8].

ALL is classified into three morphological types: L1, L2, and L3 [9]. L1 cells are the smallest, with a uniform population and coarse chromatins. L2 cells have nuclear heterogeneity and are larger than L1 cells. L3 cells have vacuoles protruding into the cells and are larger than L1 cells. Thus, early diagnosis of ALL plays a key role in recovery, especially for children [10]. The fact that the normal and lymphoid cell types have many similar characteristics, however, poses a challenge for the early diagnosis of lymphocytes. Thus, lymphocytes were classified into three types: normal, atypical, and reactive. Normal lymphocytes are characterized by homogeneity and round, small, and rough nuclei; atypical cells, by a large size and nucleus and the fact that they have lumpy chromatins; and reactive cells, by their heterogeneity and the fact that they are surrounded by red cells. Microscopic examination is the method of diagnosing the lymphocyte types; it involves taking blood or bone marrow samples, which are diagnosed by a pathologist [11]. An adequate diagnosis of leukemia, however, involves taking a bone marrow sample and analyzing it. As the analysis is made manually, it is tedious, time consuming, and sensitive to the differing opinions of experts. Therefore, an accurate manual diagnosis depends on the skill of the pathologist, although human errors may occur. Several researchers have proposed automated ways of detecting leukemia by extracting the WBC features from microscopic images. Thus, the automatic detection of blood cell images will lead to a rapid and reliable diagnosis and will allow the examination of many cells from each person. Machine and deep learning techniques can solve manual diagnostic problems. It has been proven that many of the shortcomings of manual diagnosis and medical imaging can be analyzed and solved by the convolutional neural network (CNN), which has a superior ability to distinguish between normal and blast cells. In this study, the two datasets ALL_IDB1 and ALL_IDB2 were analyzed for leukemia diagnosis using several machine and deep learning networks and hybrid techniques.

Below are the main contributions of this study:All images were optimized in both datasets by overlapping filters to obtain high-quality imagesA hybrid feature extraction technique was applied using the LBP, GLCM, and FCH algorithms, and then all the features were fused in one vector and classified using three classifiers: ANN, FFNN, and SVMThe hybrid technique was applied between CNN models to extract deep features classified using the SVM algorithm, obtaining promising diagnostic performance results.Systems were developed for analyzing blood microscopy images to assist hematologists and experts in making accurate diagnostic decisions

The rest of the paper is organized as follows. Section 2 reviews a set of relevant previous studies. Section 3 analyzes the materials and methods that were used for the processing of the images in the ALL_IDB1 and ALL_IDB2 datasets. Section 4 reviews the diagnostic performance results of the proposed systems. Section 5 discusses and compares the diagnostic performances of the proposed systems. Section 6 concludes the paper.

## 2. Related Work

This section presents methods and techniques for leukemia analysis and detection, image optimization, segmentation, classification, and manual and deep feature extraction [12]. In a study by Abhishek et al., the image optimization phase enhanced the region of interest, which affected the segmentation result [13]. Shirazi et al.’s images were improved, and the aliased and false edges were removed using the wiener filter [14]. Dhanachandra et al.’s pre-processing techniques were applied through k-means clustering, median filtering, and contrast stretching to obtain high-resolution segmented images [15]. In the study by Sell et al., the images were processed using three components of two-color spaces, CMYK and HSV, to produce new images for determining the most important features through principal component analysis to obtain WBC nuclei [16]. Saba et al. introduced the deep learning (SDL) approach to leukocyte segmentation and classification in the pre-processing and segmentation steps and conducted optimization using a generative adversarial network through its normalization, followed by deep feature extraction with the DarkNet-53 and ShuffleNet models [17]. Pouria et al. enhanced images to reduce the brightness when converting from RGB to HSV, and then applied fuzzy c-means to segment the cores and separate them from the rest of the image (the watershed transform method separates the link between the cores and the background and then extracts the most important geometric and statistical features) [18]. Angelo et al. presented an adaptive unsharpening method for improving images sorted by machine learning algorithms, and used deep learning techniques for cell diameter normalization, image improvement, focus sharpening, and diagnosis [19]. Subrajeet et al. presented a quantitative microscopic method for distinguishing malignant lymphocytes from normal blood-stained images and diagnosed lymphocytes first through segmentation using shadowed C-means clustering, and then through classification by five classification algorithms [20]. Nasir et al. discussed several phenotypic and environmental features and their feeding to four classification algorithms for diagnosing leukemia; all the algorithms obtained superior diagnosis results in all age groups [21]. Nashat et al. conducted feature selection using the particle swarm optimization algorithm with the ensemble learning method and rated the selected features using five classification algorithms; the algorithms obtained good results [22]. Supriya et al. introduced a method of diagnosing cancer cells by extracting essential features (e.g., irregularly shaped nucleus and adjacent nuclei, which indicate cancer cells) using multiple learning algorithms [23]. Israet al. presented an effective system for evaluating the blood dataset for diagnosing leukocytes, with the following stages: augmenting images, composing wavelets, and training the dataset and classifying the inputted classes using the CNN model [24]. Reena et al. presented a semantic segmentation method of isolating leukocytes from the rest of the image to extract deep feature maps using DeepLabv3 and ResNet-50; the system obtained good WBC classification results [25]. Jens et al. used a CNN model to analyze two datasets as a function of training; they trained the model, and when the images increased during the training phase, they concluded that the model was more efficient the most training datasets [26].

## 3. Materials and Methodology

This section presents the techniques, methods, and materials that were used to analyze the ALL_IDB1 and ALL_IDB2 datasets for the early detection of leukemia, as shown in Figure 1. Data were collected from patients with and without leukemia. All the images were improved, and the noise therein was removed. Then, three sets of techniques were used to analyze the improved images: (1) neural networks and machine learning algorithms based on image segmentation and hybrid feature extraction via LBP, GLCM, and FCH; (2) CNN models for extracting deep feature maps and for classifying them on the basis of the transfer learning technique; and (3) hybrid technologies consisting of two blocks each: deep learning and machine learning.

### 3.1. Description of the Two Datasets

This study evaluated neural networks, machine learning, deep learning, and hybrids on the publicly available ALL-IDB dataset containing micrographs of blood samples. The dataset focuses on ALL, the deadliest form of leukemia. All the lymphomas in each picture were identified and classified by lymphoma experts. All the images were obtained with an optical microscope with a Canon PowerShot G5 in JPG format, RGB color space (24-bit), and a high resolution (2592 × 1944). There are two types of ALL-IDB datasets: ALL_IDB1 and ALL_IDB2. The ALL_IDB1 dataset contains 108 images: 49 images of lymphomas and 59 images of normal people. Each image contains approximately 39,000 blood elements classified by lymphoma experts. The ALL-IDB2 dataset, on the other hand, contains 260 images: 130 images of lymphomas and 130 images of normal cells [27]. The ALL-IDB2 dataset is a set of regions cropped from blast cells and normal cells from the ALL-IDB1 dataset. Figure 2 shows samples from the ALL-IDB1 and ALL-IDB2 datasets (https://www.kaggle.com/nikhilsharma00/leukemia-dataset (accessed on October 2021).

### 3.2. Enhancement of Images

Preprocessing is the first step in diagnostic imaging. When blood samples are analyzed under a microscope, the microscope light is adjusted to obtain images of the samples. Thus, the brightness of the microscope varies from time to time, and the reflections also vary due to the light, all of which lead to the deterioration of the performance of artificial intelligence (AI) imaging techniques. Thus, noise reduction techniques are useful in obtaining improved images [28]. In this study, the images were enhanced by calculating the area of the RGB color channels, and then the colors were fixed through scaling. Thereafter, two filters namely, average and Laplacian filters were used to remove the noise and increase the contrast of the edges. The size of the average filter was 6 × 6. The filter was moved around the image until the image had been smoothed. The differences between the pixels were reduced by replacing each central pixel with an average of 35 adjacent pixels. Equation (1) shows the average candidate’s work.
(1)$$z\left(m\right)=\frac{1}{M}\overset{M-1}{\underset{i=0}{\sum }}y\left(m-1\right)\;$$
where *z*(*m*) is the input, *y*(*m* − 1) is the previous input, and *M* is the number of pixels in the image.

Next, a Laplacian filter that detects edges was applied. This filter detects changing regions, such as blast cells. Equation (2) describes how a Laplacian filter works.
(2)$$\nabla {\;}^{2}\;f=\frac{\partial {\;}^{2}\;f}{\partial^{\;2}\;x}+\frac{\partial {\;}^{2}\;f}{\partial^{\;2}\;y}\;$$

$\nabla {\;}^{2}\;f$ represents a differential second-order equation, and *x, y* are the coordinates of the binary matrix.

Finally, an enhanced image was obtained by subtracting the image enhanced by the Laplacian filter from the image enhanced by the average filter, as in Equation (3).
(3)$$Enhanced\;image=z\left(m\right)-\nabla {\;}^{2}\;f\;$$

Each image and its location in Figure 3 after enhancement is the same as the image with the same location in Figure 2 before enhancement.

### 3.3. Neural Networks and Machine Learning Technique

#### 3.3.1. Adaptive Region-Growing Algorithm

The images obtained under the microscope showed healthy lymphocytes and blast cells. Therefore, the analysis of complete images leads to extraction of features from healthy cells that cause a poor prognosis. The fractionation techniques take care only of the target cells and leave the unnecessary cells. In this work, the adaptive region-growing algorithm was applied. This algorithm separates similar pixels representing the pixels of lymphocytes called a region of interest (ROI) [29]. For a successful segmentation process, the following conditions must be met:

▪$\cup_{i=1}^{N}x_{\;i\;}=x$, *where m is the number of regions*▪

x=1,2,……, N is connected

▪

$P\left(x_{\;i}\right)=TRUE\;for\;1,\;2,\ldots \ldots ,N$

▪

$P\left(x_{\;i\;}\cup^{\;}x_{j}\right)FALSE\;for\;i\;\neq j,\;where\;x_{\;i\;}\;and\;x_{\;j}\;are\;neighboring\;regions$



For a successful segmentation process, the following conditions must be met: (1) the segmentation process must be complete; (2) all similar pixels must be represented in the same region; (3) specific and similar pixels must be correct when applied to other regions; and (4) no two pixels should be the same and belong to two different regions. The algorithm is based on the bottom-up approach, starting with a seed (pixel) and creating similar grouped pixels to form a region. The basic idea of the algorithm begins with many different pixels as the raw seeds for forming regions, and the regions grow gradually until all similar pixels in the same region are joined together. Figure 4 shows samples from the two datasets after the segmentation of infected and healthy cases.

#### 3.3.2. Morphological Method

The morphological method is a process for improving segmented images. After the segmentation process, holes not belonging to the ROI are left; these holes lead to a lack of diagnosis [30]. There are many morphological processes that remove these holes, such as opening, erosion, closing, and dilation. Morphological processes create a structure element with a 5 × 5 size, move it to each location of the segmentation image, and compare it with the neighboring pixels. The processes test if the structure element “fits in” with its neighbors or not, and the “hits” process tests the intersection between neighbors. The process continues until the whole image is scanned and an optimized binary image is produced. Figure 5 shows samples of dataset images obtained before and after the application of morphological processes. Where it is noted, the images were improved by filling in the gaps not belonging to the ROI.

#### 3.3.3. Feature Extraction

As medical images contain thousands of data that are difficult to analyze with high efficiency, feature extraction is one of the essential stages of medical image processing because it reduces the dimensions of the original image and extracts the essential representative features only from the ROI area [31]. In this work, features from the ROI were extracted using three algorithms: LBP, GLCM, and FCH. Thereafter, all the features extracted through these three methods were combined to produce robust representative feature vectors. The feature fusion method is a modern method whose purpose is to obtain more significant and representative features for high-performance diagnostic systems.

The LBP algorithm is a feature extraction method that works by selecting the central pixel and the pixels adjacent thereto [31]. The algorithm is set to a 5 × 5 size. In each iteration, the algorithm selects the central pixel (*g_c_*), its neighboring pixels (*g_p_*), and the radius R, which is 24 neighboring pixels for each central pixel. Equations (4) and (5) describe how the LBP algorithm works by replacing each central pixel according to the algorithm equation with 24 adjacent pixels. The process is repeated until all the pixels of the image are covered. A total of 203 representative features were extracted from each image using this method, which were stored in a feature vector [32].
(4)$$LBP\;{\left(x_{c},\;y_{c}\right)}_{R,P}=\overset{P-1}{\underset{P=0}{\sum }}s\left((g_{p}-g_{c}\right).2^{P}$$
(5)$$x\left(c\right)=\{\begin{array}{c} 0,\;\;c<0\\ 1,\;\;c\geq 0 \end{array}$$
where *P* is the number of pixels in the whole image and *R* is the number of neighboring pixels for each central pixel.

The GLCM algorithm is one of the best algorithms for extracting texture features from the ROI. It shows the multiple synthetic grey levels in the ROI Additionally, it calculates the statistical features according to the GLCM. The algorithm relies on spatial information to distinguish between smooth pixels with close pixel values and coarse pixels with divergent pixel values. Spatial information is important for determining the correlation between pairs of pixels in terms of distance d and the directions between pixels θ that determine the location of the first pixel from the second. The directions are selected on the basis of four directions: 0°, 45°, 90°, and 135°. The value of d is 1 when the angle θ between the pairs of pixels is 0 or 90, while the value of d is √2 when the angle θ between the pairs of pixels is 45 or 135 [33]. Thirteen representative statistical features were extracted from each image using this method, all of which were stored in a feature vector.

The FCH algorithm is one of the most important algorithms for extracting color features from the ROI. Color is a vital feature for lymphocyte diagnosis, and each color is represented in the histogram bin. Thus, each color in the ROI represents several histogram bins [31]. The algorithm deals with two similar colors in the same histogram bin. Even if two colors are almost the same, they are considered different when they are in different histogram bins. Thus, the algorithm uses the membership value of each pixel to check for color similarity. The colors of a histogram in the ROI are considered *I* containing n pixels as $X\left(I\right)=x_{1,}x_{2,}\ldots \ldots \ldots x_{i}$, where *n_i_* refers to the pixels of the image, $i^{th}\;$ refers to all the color bins, and $x_{i=ni/n}$ indicates the probability that any image gain belongs to several color bins.
(6)$$x_{i}=\overset{n}{\underset{j=1}{\sum }}p_{i/j}\;p_{j\;\;=\;}\frac{1}{n}\;\overset{n}{\underset{j=1}{\sum }}p_{i/j}\;$$
where *p_j_* refers to image pixels converted into conditional probability $p_{i/j}$, which means the probability of the $j^{th}$ pixel belonging to *I* color bins using the FCH algorithm.
(7)$$p_{i/j}=\{\begin{array}{c} 1,\;\;\;\;\;\;\;\;\;\mathrm{if}\;\;\;j^{th}\;\;\mathrm{pixel}\;\;\mathrm{belongs}\;\mathrm{to}\;\mathrm{the}\;i^{th}\;\mathrm{color}\;\mathrm{bin}\\ 0,\;\;\;\;\;\;\;\;\;\;\;\;\;\;\;\;\;\;\;\;\;\;\;\;\;\;\;\;\;\;\;\;\;\;\;\;\;\;\;\;\;\;\;\;\;\;\;\;\;\;\;\;\mathrm{Otherwise} \end{array}\;$$

Sixteen color features were extracted from each image using the aforementioned method, all of which were stored in a feature vector.

Finally, the features extracted from the three aforementioned methods were merged or fused into one feature vector for each image to obtain highly efficient representative features. The LBP algorithm produces 203 features; the GLCM algorithm, 13; and the FCH algorithm, 16. Thus, when all the features are combined in one vector, the number of representative features becomes 232 for each image. Figure 6 shows the process of fusion of all the features extracted by the LBP, GLCM and FCH algorithms.

#### 3.3.4. Classification

##### Artificial Neural Network and Feed Forward Neural Network Algorithms

In this section, the ANN, FFNN, and SVM algorithms are used to evaluate the ALL-IDB dataset. ANN is a type of intelligent network consisting of many neurons interconnected with each other and with other layers. It has the ability to obtain, analyze, and interpret information from raw and complex data. It also has the ability to adapt to any data in any circumstance and reduces the possibility of errors for any entry belonging to any class. Data pass from the ANN, flow between neurons and layers, and are stored between neurons as points called weights. ANNs consist of an input layer that receives the features extracted from the previous stage. Many complex mathematical operations are performed to carry out the required tasks and to solve the problems in many of the hidden layers. The output layer consists of many neurons (classes) [34]. The network is characterized by many features as a unit for processing the required tasks and the activation function, by weights connecting neurons, and by a spread bias that activates neurons with external inputs. It is also characterized by the rule of learning and calculating the error signals through the minimum sum squared, which measures the error rate between the actual and predicted outputs, as shown in Equation (8). The algorithm continuously updates the weights until the minimum square error is obtained between the actual output X and the predicted output Y.
(8)$$\;\mathrm{MSE}=\frac{1}{n}\overset{n}{\underset{i=1}{\sum }}\;{\left(\;X_{i}-Y_{i}\right)}^{2}$$

In this work, the ANN algorithm consisted of an input layer containing 232 neurons representing the features extracted from the previous stage. There were 10 hidden layers for handling all the required tasks. Each hidden layer had many neurons interconnected with specific weights. The output layer, which contained two neurons, indicated if the case was a leukemia case or was a normal case. Figure 7 shows the architecture of the ANN algorithm for evaluating the ALL-IDB dataset.

The FFNN algorithm is a computational network used to solve complex problems. Its architecture is similar to that of the ANN algorithm, consisting of input, hidden, and output layers. For the algorithm’s mechanism, the input layer receives 232 inputs (features) and the hidden layer is fed with the outputs of the input layer. The hidden layer contains many neurons interconnected by connections called weights (w). The network contains 10 hidden layers. Information flows between the neurons in a forward direction. Each neuron produces an output on the basis of its weight multiplied by the output of the previous neurons. The weights are updated and modified from the hidden layer to the output layer until a minimum sum squared is obtained. The minimum sum squared is calculated on the basis of the MSE equation above, representing the difference between the actual and predicted outputs.

##### Support Vector Machine Algorithm

The SVM algorithm is a machine learning algorithm used to solve classification and regression problems. The algorithm plots each data element as a point in an n-dimensional space (*n* is the number of features to be extracted). The points are then categorized into two separate classes by hyperplane. Hyperplanes are what help classify the data and are the decision boundaries of the algorithm. All data points on either side of a hyperplane are different data points, where all the points above the hyperplane are classified as Class 1 and all the points below the hyperplane are classified as Class 2. At the same time, the data points located on or near a hyperplane are called support vectors. Support vectors are used to maximize the margin of the algorithm. Changing or deleting support vectors causes hyperplanes to change, so these points help build an SVM classifier. The algorithm produces multiple hyperplanes, and the hyperplanes with a maximizing margin are selected. The margin between points is maximized through a loss function that helps maximize the margin between hyperplanes and data points.

### 3.4. Convolutional Neural Networks (CNNs)

Deep learning is a machine learning technique in which many layers of information processing units are used in the unsupervised learning of characteristics and for the analysis or classification of patterns (supervised learning) [35]. The essence of deep learning is to obtain multiple levels of representation, from simple nonlinear modules that transform the representation from one level to a higher and more abstract one. For classification purposes, the higher representation layers amplify the aspects of the input that are most important for discriminating between classes and suppress the variations that are irrelevant. Architecture depth refers to the number of levels of nonlinear operations learned. As the algorithms commonly used in machine learning correspond to superficial architectures, ANN researchers have made efforts to replicate this type of architecture [36]. Deep neural networks have been successfully applied in classification, regression, dimension reduction, texture modeling, motion modeling, object segmentation, medical image classification, robotics, natural language processing, image recognition, speech recognition, signal processing, and others. A CNN consists of many layers: the convolutional layer, the max- or average-pooling layer, the fully connected layer (FCL), and other auxiliary layers.

One of the most critical layers is the convolutional layer, which gives CNNs their name. This layer performs a linear operation called convolution between filter *w(t)* and image *x(t),* and writes *(x*w)(t)* or *s(t),* as in Equation (9). There are three parameters that control the convolution layer: filter size, zero padding, and p-step. The larger the filter size, the larger the wrapping around the images. Each filter is designed to detect specific features in the input image. For example, a filter is designed to detect edges, another is designed to detect geometric features, and another is designed to detect textures and colors. Thus, this capability of CNNs is called translation invariance. Zero padding is used to maintain the size of the original input. The size of the zero pad is determined on the basis of the sizes of the convolutional filter and original input. The p-step parameter is used to determine the number of steps taken by the filter on the image at a time.
(9)$$s\left(t\right)=\left(x*w\right)\left(t\right)=\int^{\;}x\left(a\right)w\left(t-a\right)\;\;da$$
where *s*(*t*) represents the convolution output, which produces a feature map. If *t* and *w* are both integer values, the convolution is represented by Equation (10).
(10)$$s\left(t\right)=\left(x*w\right)\left(t\right)=\overset{\infty }{\underset{-\infty }{\sum }}x\left(a\right)w\left(t-a\right)\;\;$$

In CNN implementation, emphasis should be placed on an image’s dimensions and color spaces; thus, convolutional filters are adapted to the input images. In the case of two-dimensional (2D) images, the convolutional layer of filter *K* with input image *I* as shown in Equation (11).
(11)$$s\left(i,j\right)=\left(I*K\right)\left(i,j\right)=\underset{m}{\sum }\underset{n}{\sum }I\left(m,n\right)\;K\left(i-m,\;j-n\right),\;\;$$

In the case of the input RGB images, the convolutional layers work on 2D convolutions for each color: one for color R, one for color G, and one for color B. Several convolutional layers are followed by a rectified linear unit (ReLU) layer for further processing. This layer passes the positive input and suppresses the negative input and converts it into 0. Equation (12) describes how the ReLU layer passes only positive values and converts negative values into 0.
(12)$$\mathrm{ReLU}\left(x\right)=\max\left(\;0,\;x\;\right)=\{\begin{array}{l} x,\;\;x\geq 0\\ 0,\;\;x<0 \end{array}$$

As convolutional layers produce millions of parameters, an overfitting problem occurs. CNNs provide a solution to this problem by using a dropout layer. The dropout layer stops 50% of the neurons and pass 50% on each iteration, and the process continues. In this study, the dropout layer was set to 50%. However, this layer doubles the training time.

Convolutional layers produce high-dimensional feature maps that slow down the training process. Thus, to speed up the training process, CNNs provide pooling layers to reduce the dimensions. Pooling layers interact within CNNs with the same convolutional layer mechanism. There are two types of pooling layers: the max- and average-pooling layers. In the max-pooling layer work mechanism, the maximum value is chosen from the specified values, as shown in Equation (13). In the average-pooling layer work mechanism, on the other hand, the average specified values are calculated and replaced by the average value, as shown in Equation (14).
(13)$$P\left(i;\;j\right)=max_{m,n=1\ldots .k}\;A\left[\left(i-1\right)p+m;\;\left(j-1\right)p+n\right]\;\;$$
(14)$$P\left(i;\;j\right)=\frac{1}{k^{2}}\underset{m,n=1\ldots .k}{\sum }A\left[\left(i-1\right)p+m;\;\left(j-1\right)p+n\right]\;\;$$
where *A* is the pooling layers’ filter size; *m, n* are the filter size dimensions; *p* is the filter step size; and *k* is the filter capacity.

The FCL is the layer responsible for classification in CNNs. The FCL is characterized by the interconnection of all neurons. The FCL layer converts 2D deep feature maps into one-dimensional maps. The number of FCLs varies from one CNN to another; some networks have more than one FCL, which classify each image into the correct class. Finally, the FCL output is fed to the softmax activation function to produce neurons with the same number of classes entered. In this study, softmax produced two neurons (classes): leukemia and normal. Equation (15) shows how the softmax function works.
(15)$$y\left(x_{i}\right)=\frac{\exp x_{i}}{\;\sum_{j=1}^{n}\;\exp x_{j}}\;$$
where *y*(*x*) is softmax, which is between 0 *≤ y*(*x*) *≤* 1.

This study focused on three CNN models: AlexNet, GoogLeNet, and ResNet-18.

#### 3.4.1. AlexNet Model

AlexNet is a CNN containing 25 layers divided into many layers from the input of images to the final classification. The most important layers are five convolutional layers, several ReLU layers, three max-pooling layers, two dropout layers, three FCLs, and a softmax layer [37]. AlexNet also contains 650,000 neurons, 62 million parameters, and 630 million connections between neurons. Figure 8 shows the AlexNet architecture and the most critical layers it contains for analyzing the ALL-IDB dataset and classifying the data into two classes: leukemia and normal.

#### 3.4.2. GoogLeNet Model

GoogLeNet is a CNN used for pattern recognition and other computer vision applications. It has 27 layers, including pooling layers. The model is characterized by its significant reduction of image dimensions, high computational efficiency, and ability to preserve important information [38]. The convolutional layer contains a filter with a 7 × 7 size, a large size that reduces the image dimensions dramatically as it represents 49 pixels by one pixel. GoogLeNet works to reduce the image dimensions and reduces the image height and width through three max-pooling layers 3 × 3 in size, in addition to one with a 7 × 7-layer size, which significantly reduces the dimensions of the image. The network also produces more than seven million barometers. Figure 9 shows the GoogLeNet architecture for analyzing the ALL-IDB dataset and classifying the data into either leukemia or normal classes.

#### 3.4.3. ResNet-18 Model

ResNet-18 is a CNN belonging to the ResNet-xx family of models. ResNet-18 contains 18 layers: five convolutional layers to produce feature maps, a ReLU layer, an average-pooling layer for dimensionality reduction, an FCL, a softmax activation function that classifies the ALL-IDB dataset images into two classes (leukemia or normal), and many auxiliary layers [39]. Figure 10 shows the infrastructure of the ResNet-18 network, which produces more than 11.5 million parameters.

### 3.5. Deep Learning–Machine Learning Hybrid Techniques

In this section, we present new techniques: hybrid techniques between machine learning and deep learning for evaluating the ALL-IDB dataset for the early detection of leukemia. These hybrid techniques are proposed because the use of deep learning networks poses some challenges, such as the requirement of high-specification computers and the time-consuming process of training the dataset [40]. Thus, the hybrid techniques require medium-specification computers, and the process of training the dataset is fast and not time consuming. In this study, hybrid techniques consisting of two blocks each were used. The first block consisted of the CNN models namely AlexNet, GoogleNet, and ResNet-18, which extract deep features and feed them into the second block. The second block was the SVM classification algorithm, which classifies the deep feature maps extracted from the CNNs. Figure 11a–c show the hybrid architecture consisting of two blocks: deep learning and machine learning (called AlexNet + SVM and GoogleNet + SVM, respectively) and ResNet-18 + SVM. It can be seen in the figure that the FCLs in the CNNs (the first block) have been replaced by the SVM classifier (the second block).

## 4. Experimental Results

### 4.1. Dataset Splitting and Environment Setup

All the systems proposed in this study were implemented on the ALL_IDB1 and ALL_IDB2 datasets. The ALL_IDB1 dataset contains 108 images divided into two classes: leukemia (49 images; 45.37%) and normal (59 images; 54.63%). The ALL_IDB2 dataset, on the other hand, contains 260 images equally divided between the leukemia and normal classes (130 images per class). The two datasets were divided into 80% for training and validation (80:20%) and 20% for testing. Table 1 shows the splitting of the ALL_IDB1 and ALL_IDB2 datasets during the training, validation, and testing phases for both the leukemia and normal classes. All the proposed systems were implemented with the Intel^®^ i5 processor, 12 GB RAM, and GPU 4 GB GEFORCE computer specifications and the MATLAB 2018b environment.

### 4.2. Evaluation Metrics

In this section, we describe the statistical metrics that were used to evaluate the diagnostic performances of the proposed systems (the ANN, FFNN, and SVM techniques; the CNN models AlexNet, GoogleNet, and ResNet-18; and the hybrid techniques AlexNet + SVM, GoogleNet + SVM, and ResNet-18 + SVM) in the ALL_IDB1 and ALL_IDB2 datasets, or to determine if they were effective or not. The metrics that were used were accuracy, precision, sensitivity, specificity, and area under the curve (AUC), described in Equations (16)–(20), respectively. The equations’ information was obtained through the confusion matrix, which was in turn obtained from the outputs of the proposed systems. The confusion matrix contained the classification information for all the inputted images. There were correctly rated images called TP and TN and incorrectly rated images called FP and FN [41].
(16)$$\mathrm{Accuracy}=\frac{\mathrm{TN}+\mathrm{TP}}{\mathrm{TN}+\mathrm{TP}+\mathrm{FN}+\mathrm{FP}}\;\;*100\%\;$$
(17)$$\mathrm{Precision}=\frac{\mathrm{TP}}{\mathrm{TP}+\mathrm{FP}}\;\;*100\%$$
(18)$$\mathrm{Sensitivity}=\frac{\mathrm{TP}}{\mathrm{TP}+\mathrm{FN}}\;\;*100\%\;$$
(19)$$\mathrm{Specificity}=\frac{\mathrm{TN}}{\mathrm{TN}+\mathrm{FP}}\;\;*100$$
(20)$$\mathrm{AUC}\;=\frac{\mathrm{True}\;\mathrm{Positive}\;\mathrm{Rate}}{\mathrm{False}\;\mathrm{Positive}\;\mathrm{Rate}}=\frac{\mathrm{Sensitivity}}{\mathrm{Specificity}}$$
where TP is the number of correctly classified leukemia patients, TN is the number of correctly classified normal patients, FN is the number of leukemia patients wrongly classified as normal patients, and FP is the number of normal patients wrongly classified as leukemia patients.

### 4.3. Results of the ANN, FFNN and SVM Algorithms

Neural networks algorithms are among the most efficient algorithms for diagnostic imaging. Such algorithms depend on performance accuracy in the previous stages of image processing: preprocessing, segmentation, and feature extraction. As shown above, the dataset was divided into a training and validation set and a system proficiency testing set. Figure 12 shows the training of the dataset for the ANN and FFNN algorithms, with the input layer containing the input neurons (232 extracted features or neurons) and 10 hidden layers for carrying out all the required tasks and calculations for diagnosing leukemia. The output layer consisted of two neurons: leukemia and normal. We will discuss the diagnostic performances of the ANN, FFNN, and SVM algorithms in detail.

#### 4.3.1. Performance Analysis

One measure of the performances of the ANN and FFNN algorithms is cross-entropy, which measures the error rate between the actual and predicted outputs. Figure 13 shows the performance of the ANN algorithm for the ALL_IDB1 and ALL_IDB2 datasets during the training, validation, and testing phases. The red color depicts the algorithm’s performance during the testing phase; the blue color, during the training phase; and the green color, during the validation phase. The best performance was extracted through the intersecting lines. The error rate between the actual and predicted outputs decreased as the epochs increased; the training stopped when the algorithm reached the minimum error value. It can be seen in Figure 13a that for the ALL_IDB1 dataset, the ANN algorithm achieved the best validation performance of 0.043769 during epoch 5, and that all the stages had the same performance. As can be seen in Figure 13b, on the other hand, for the ALL_IDB2 dataset, the ANN algorithm achieved the best validation performance of 0.006686 during epoch 7.

#### 4.3.2. Gradient

Gradient values and validation check are among the performance measures of the ANN and FFNN algorithms, measuring the error rates. Figure 14 shows the performance of the ANN algorithm for the ALL_IDB1 and ALL_IDB2 datasets during the training phase. It can be noted in Figure 14a that for the ALL_IDB1 dataset, the ANN algorithm achieved the minimum error value at a 0.3916 gradient value at epoch 1000 and a validation check value of 2 at epoch 1000. As can be seen in Figure 14b, on the other hand, for the ALL_IDB2 dataset, the ANN algorithm achieved the minimum error value at a 0.046368 gradient value at epoch 53 and a validation check value of 6 at epoch 53.

#### 4.3.3. Receiver Operating Characteristic (ROC)

ROC is an effective measure of the performances of classification algorithms such as ANN and FFNN. ROC is also called area under the curve (AUC). An algorithm achieves the best AUC when the curve approaches the left corner. AUC is calculated by dividing the sensitivity value by the specificity level, where sensitivity represents the y-axis called true positive rate and specificity represents the x-axis called false positive rate. Figure 15 shows the performance of the ANN algorithm for the ALL_IDB1 and ALL_IDB2 datasets during the training, validation, testing, and overall phases. It can be seen in Figure 15a that for the ALL_IDB1 dataset, the performance of the ANN algorithm achieved an overall average AUC of 94.52%, whereas it can be seen in Figure 15b that for the ALL_IDB2 dataset, the ANN algorithm achieved an overall AUC ratio of 99.21%.

#### 4.3.4. Error Histogram

The error histogram measures the performances of classification algorithms such as ANN and FFNN. It evaluates an algorithm’s performance to reach the minimum error between the actual and expected outputs represented by the x-axis. Errors occurring during the training phase are represented by the blue histogram bin whereas those occurring during the validation and testing phases are represented by the red histogram bin. A dataset may contain outliers, so the histogram bin provides information about the outliers or the values that are behaving differently from the original values. Figure 16 shows the performance of the ANN algorithm for the ALL_IDB1 and ALL_IDB2 datasets during the training, validation, and testing phases. It can be seen in Figure 16a that the performance of the ANN algorithm for the ALL_IDB1 dataset reached the minimum error value between the actual and expected outputs, with 20 bins between −0.8142 and 1.204, whereas in Figure 16b, for the ALL_IDB2 dataset, the ANN algorithm reached the minimum error value between the actual and predicted outputs, with 20 bins between −0.5247 and 1.091.

#### 4.3.5. Regression

Regression is one of the performance measures of neural networks algorithms. It predicts a continuous variable on the basis of the other variables. Algorithms predict the predicted output on the basis of the actual output, by comparing the actual and expected outputs. When the value of R is close to 1, this means that the prediction of the predicted output based on the actual output has reached the minimum error value, and the relationship between the predicted and actual outputs is strong. Figure 17 shows the performance regression of the FFNN algorithm for the ALL_IDB1 and ALL_IDB2 datasets during the training, validation, and testing phases. It can be seen in Figure 17a that the performance of the FFNN algorithm for the ALL_IDB1 dataset reached the minimum error value between the actual and predicted outputs, and the global regression value reached 0.98899. On the other hand, in Figure 17b, the performance of the FFNN algorithm for the ALL_IDB2 AUC dataset reached the minimum error value between the actual and expected outputs, and the overall regression value reached 1. This means that the actual output–predicted output relationship was 100% and reached the minimum error value 0.

#### 4.3.6. Confusion Matrix

The confusion matrix is the most critical measure of the performances of all classifiers and networks, describing all the images that enter the system. It is the output of the algorithm and is similar to a matrix with rows and columns. Each row is a class that represents the correctly categorized images in the primary diameter cell, and the rest of the cells are considered incorrectly categorized images. The confusion matrix contains incorrectly rated images called TP and TN, and incorrectly rated images called FP and FN. The FFNN and ANN algorithms produced a confusion matrix for the ALL_IDB1 and ALL_IDB2 datasets, where Class 1 represents images of abnormal blood samples (leukemia) and Class 2 represents images of normal blood samples. Figure 18 describes the confusion matrix ANN for the ALL_IDB1 and ALL_IDB2 datasets. It can be seen in Figure 18a that the performance of the ANN algorithm for the ALL_IDB1 dataset achieved only 94.4% accuracy whereas in Figure 18b, the performance of the ANN algorithm for the ALL_IDB2 dataset achieved 100% accuracy. Figure 19 shows the confusion matrix of the FFNN algorithm for the ALL_IDB1 and ALL_IDB2 datasets. It can be seen in Figure 19a,b, respectively, that the performance of the FFNN algorithm for the ALL_IDB1 dataset and the performance of the FFNN algorithm for the ALL_IDB2 dataset both achieved 100% accuracy.

Table 2 summarizes the performance results of the FFNN, ANN, and SVM algorithms for the ALL_IDB1 and ALL_IDB2 datasets for the early detection of leukemia. It was noted that the FFNN algorithm is superior to the rest of the algorithms in both the ALL_IDB1 and ALL_IDB2 datasets, achieving 100% accuracy for all the measures. The ANN algorithm achieved 94.4% accuracy, 100% precision, 91.55% sensitivity, 91.55% specificity, and 94.52% AUC for the ALL_IDB1 dataset, whereas for the ALL_IDB2 dataset, it achieved 100% accuracy, 100% precision, 100% sensitivity, 100% specificity, and 99.21% AUC. The SVM algorithm, on the other hand, achieved 90.91% accuracy, 100% precision, 84.62% sensitivity, 100% specificity, and 91.99% AUC for the ALL_IDB1 dataset, whereas for the ALL_IDB2 dataset, it achieved 98.11% accuracy, 95.83% precision, 100% sensitivity, 96.67% specificity, and 97.86% AUC.

Figure 20 describes the performances of all the neural networks algorithms (ANN, FFNN, and SVM) for the ALL_IDB1 and ALL_IDB2 datasets in a graph.

### 4.4. Results of the Convolutional Neural Network Models

In this section, the performances of the three CNN models AlexNet, GoogLeNet, and ResNet-18 for the ALL_IDB1 and ALL_IDB2 datasets using the transfer learning technique is discussed. The models were trained on millions of images for classification into more than a thousand classes, and then the transfer learning technique was used to transfer the acquired experience to the performance of new tasks for the ALL_IDB1 and ALL_IDB2 datasets. Both datasets contain few images, affecting the diagnostic accuracy because CNN models require a large dataset. The CNN models address this challenge by applying the data augmentation technique. This technique was used for the AlexNet, GoogLeNet, and ResNet-18 networks. The images were artificially augmented from the same dataset, and many operations were applied, such as image rotation at several angles, flipping, cropping, and displacement.

Table 3 summarizes the sizes of the two datasets before and after the application of the data augmentation method to obtain a sufficient number of images for training the two datasets and to solve the problem of overfitting and balancing the datasets. An increase in the images was noted during the training phase, with the images for the leukemia class in the ALL_IDB1 dataset being increased 20 times. On the other hand, the images for the normal class were increased 17 times to obtain a balanced dataset during the training phase. For the ALL_IDB2 dataset, the images for the two classes were increased 10 times.

Table 4 shows the tuning of the CNNs AlexNet, GoogLeNet, and ResNet-18, where the optimizer was set to adam, Mini Batch Size, Max Epochs, Validation Frequency, Initial Learn Rate, and Execution Environment.

The parameters of the CNN models were set by many experiments. In each experiment, the results of the performance of the CNN models were recorded until the best performance of the CNN models was reached when setting the parameters shown in the table.

Table 5 shows the performance results of the CNNs AlexNet, GoogLeNet, and ResNet-18. The GoogLeNet model outperformed the rest of the models in evaluating the ALL_IDB1 and ALL_IDB2 datasets for leukemia detection, achieving 100% accuracy, precision, sensitivity, specificity, and AUC. The AlexNet model obtained a score of 100% for all the metrics for the ALL_IDB1 dataset, whereas for the ALL_IDB2 dataset, it achieved 94.2% accuracy, 92.3% precision, 96.2% sensitivity, 94.5% specificity, and 99.26% AUC. As for the ResNet-18 model, it obtained a score of 100% for all the metrics for the ALL_IDB1 dataset, whereas for the ALL_IDB2 dataset, it achieved 97.44% accuracy, 97.4% precision, 97.4% sensitivity, 97.4% specificity, and 99.93% AUC.

Figure 21 shows the performances of all the CNN models for the ALL_IDB1 and ALL_IDB2 datasets in a graph.

Figure 22 shows the confusion matrix representing the AlexNet model for the early detection of leukemia from the ALL_IDB1 and ALL_IDB2 datasets. The confusion matrix contains all the dataset samples correctly labeled in the primary diameter (TP and TN) and incorrectly classified in the secondary diameter (FP and FN). Figure 22a shows that the AlexNet model achieved an overall accuracy rate of 100% in each class diagnosis for the ALL_IDB1 dataset, whereas Figure 22b shows that the same model achieved 94.2% overall accuracy, 96.2% leukemia diagnosis accuracy, and 92.3% normal-sample diagnosis accuracy for the ALL_IDB2 dataset.

Shown in Figure 23 is a confusion matrix representing the output of the GoogleNet model for the early detection of leukemia from the ALL_IDB1 and ALL_IDB2 datasets. Figure 23a,b show that the GoogleNet model reached 100% overall accuracy for each class diagnosis for both the ALL_IDB1 and ALL_IDB2 datasets.

Figure 24 shows the confusion matrix representing the ResNet-18 model for the early detection of leukemia from the ALL_IDB1 and ALL_IDB2 datasets. Figure 24a shows that the ResNet-18 model achieved 100% overall accuracy of each class diagnosis from the ALL_IDB1 dataset, whereas Figure 24b shows that the same model achieved 97.4% overall accuracy, 97.4% leukemia diagnosis accuracy, and 97.4% normal-sample diagnosis accuracy from the ALL_IDB2 dataset.

### 4.5. Results of the Hybrid Convolutional Neural Network Models with the Support Vector Machine Classifier

In this section, the results of the new CNN (AlexNet, GoogleNet, and ResNet-18)–machine learning classifier (SVM) hybrid techniques are presented. One of the reasons for applying these hybrid techniques is their flexibility with regard to computer resources, as they require only medium-specification computers, unlike CNN models, which require high-specification computers. In addition, hybrid techniques are characterized by their speed during training, unlike CNN models, which consume much time for training. Thus, hybrid techniques consisting of two blocks were applied: (1) CNNs (AlexNet, GoogleNet, and ResNet-18), with the task of extracting deep feature maps; and (2) an SVM classifier, with the task of classifying deep feature maps with high accuracy and speed. Thus, these hybrid techniques were AlexNet + SVM, GoogleNet + SVM, and ResNet-18 + SVM.

Table 6 shows the performance results of the AlexNet + SVM, GoogleNet + SVM, and ResNet-18 + SVM hybrid techniques. It can be seen that ResNet-18 + SVM achieved better results than the other networks for both the ALL_IDB1 and ALL_IDB2 datasets (100% accuracy, precision, sensitivity, specificity, and AUC). AlexNet + SVM achieved a score of 100% for all the metrics for the ALL_IDB1 dataset, whereas for the ALL_IDB2 dataset, it achieved 96.2% accuracy, 96.2% precision, 96.2% sensitivity, 96.2% specificity, and 98.56% AUC. GoogLeNet + SVM achieved 95.5% accuracy, 95.45% precision, 100% sensitivity, 91.7% specificity, and 99.34% AUC for the ALL_IDB1 dataset and 98.1% accuracy, 98% precision, 96.2% sensitivity, 100% specificity, and 99.87% AUC for the ALL_IDB2 dataset.

Figure 25 shows the performance of all the hybrid techniques for the ALL_IDB1 and ALL_IDB2 datasets in a graph.

Figure 26 shows the confusion matrix generated by the AlexNet + SVM hybrid technique for the early detection of leukemia from the ALL_IDB1 and ALL_IDB2 datasets. Figure 26a shows that the AlexNet + SVM model achieved 100% overall accuracy of each category diagnosis from the ALL_IDB1 dataset, whereas Figure 26b shows that the same model achieved 96.2% overall accuracy, 96.2% leukemia diagnosis accuracy, and 96.2% normal-sample diagnosis accuracy for the ALL_IDB2 dataset.

The confusion matrix shown in Figure 27 represents the performance of the GoogLeNet + SVM hybrid network for the early detection of leukemia from the ALL_IDB1 and ALL_IDB2 datasets. Figure 27a shows that the GoogLeNet + SVM network achieved 95.5% overall accuracy, 100% leukemia sample diagnosis accuracy, and 91.7% normal-sample diagnosis accuracy, whereas Figure 27b shows that the same network achieved 98.1% overall accuracy, 96.2% leukemia diagnosis accuracy, and 100% normal-sample diagnosis accuracy for the ALL_IDB2 dataset.

Figure 28 shows the confusion matrix of the ResNet-18 + SVM hybrid technique for the early detection of leukemia from the ALL_IDB1 and ALL_IDB2 datasets. Figure 28a,b show that the ResNet-18 + SVM model achieved 100% overall superior accuracy for each class for both the ALL_IDB1 and ALL_IDB2 datasets.

## 5. Discussion and Comparison with Previous Studies

In this section, we discuss the proposed techniques, which were developed for the early detection of leukemia from the ALL_IDB1 and ALL_IDB2 datasets: neural networks and machine learning algorithms (ANN, FFNN, and SVM), CNN models (AlexNet, GoogLeNet, and ResNet-18), and CNN–SVM hybrid techniques (AlexNet + SVM, GoogLeNet + SVM, and ResNet-18 + SVM). All the images were optimized for the two datasets, and all the proposed systems were optimized using the same technique. The two datasets were divided into 80% for training and 20% for testing, taking into account the data augmentation technique during the training phase of the CNN model (AlexNet, GoogLeNet, and ResNet-18).

In the first proposed system, lesion segmentation was caried out, the ROI was obtained, and the most important features were extracted by the LBP, GLCM, and FCH algorithms. The three methods’ features were combined into one vector, which produced 232 characteristics for each image, and hybrid features were fed into ANN, FFNN, and SVM classifiers. All the features were classified as either leukemia or normal. All the algorithms achieved superior accuracy for both the ALL_IDB1 and ALL_IDB2 datasets.

In the second proposed system consisting of CNNs (AlexNet, GoogLeNet, and ResNet-18), for the ALL_IDB1 dataset, all the models achieved 100% scores for all the metrics. For the ALL_IDB2 dataset, the GoogLeNet model achieved better results than the other models (100% for all the metrics).

In the third proposed system consisting of hybrid technologies (AlexNet + SVM, GoogLeNet + SVM, and ResNet-18 + SVM), feature maps were extracted from CNN models and were classified by an SVM classifier. For the ALL_IDB1 dataset, AlexNet + SVM and ResNet-18 + SVM achieved superior results (100% for all the metrics). For the ALL_IDB2 dataset, all the networks achieved superior results, but ResNet-18 + SVM outperformed the rest of the networks, achieving a score of 100% for all the metrics.

Table 7 summarizes the levels of diagnostic accuracy achieved by all the proposed systems for the ALL_IDB1 and ALL_IDB2 datasets. For the ALL_IDB1 dataset, the best leukemia diagnostic accuracy was obtained by the FFNN, AlexNet, GoogLeNet, ResNet-18, AlexNet + SVM, GoogLeNet + SVM, and ResNet-18 + SVM networks, all of which achieved 100% accuracy. For the ALL_IDB2 dataset, the best leukemia diagnostic accuracy was obtained by the ANN, FFNN, SVM, GoogleNet, and ResNet-18 + SVM networks, which achieved 100% accuracy. Figure 29 compares the performances of all the proposed systems in diagnosing leukemia of the two datasets.

Table 8 summarizes the evaluation results of previous systems compared to the performance of our proposed relevant systems. It is noted that the accuracy reached by the previous systems ranged between 89.40% and 93.7%, while the accuracy of our proposed system reached 100%. As for precision, the previous systems reached 90.10% and 94.01%, while our proposed system reached 100%. Regarding the sensitivity, the previous systems reached between 89.40% and 93.7%, while the sensitivity of our proposed system reached 100%. As for the specificity, the previous systems reached 87.50% and 98.4%, while our proposed system reached 100%. Finally, the previous systems reached an AUC of 83.20% and 97.46%, while our system achieved 100%.

Figure 30 display the performance comparison of our proposed system with previous related systems.

## 6. Conclusions and Future Work

AI algorithms have emerged in the medical sector to provide powerful analytical and diagnostic tools with high efficiency. Machine and deep learning techniques address the challenges of shortcomings in manual diagnosis, differing opinions of experts, and the time-consuming tracking of blood samples. These techniques play a key role in the early detection of leukemia. This study presented many of the proposed systems for the ALL_IDB1 and ALL_IDB2 datasets, distributed as follows. The first set of proposed systems consist of the three algorithms ANN, FFNN, and SVM, whose classification is based on the algorithms extracted in a hybrid way by LBP, GLGM, and FCH; all the systems achieved superior results for both datasets. The second set of proposed systems consists of the three CNN models AlexNet, GoogLeNet, and ResNet-18, which categorize the feature maps extracted by the FCL on the basis of the transfer learning method. All the models achieved promising results for both datasets. The third set of proposed systems consists of the three hybrid CNN–SVM technologies AlexNet + SVM, GoogLeNet + SVM, and ResNet-18 + SVM, each of which consists of two blocks: (1) CNN models for extracting deep feature maps and (2) the SVM algorithm for classifying the feature maps extracted from the first block. All the hybrid techniques obtained superior results for both datasets.

CNN models require a large data set to avoid overfitting, and this is one of the limitations in this study because the data set is insufficient to train CNN models. This limitation was overcome by using the data augmentation technique.

Future work of the proposed systems would involve applying a hybrid technique between the features extracted by CNN models and combining them with the features extracted by the LBP, GLCM and FCH algorithms in feature vector; then, feeding them into classifiers of neural networks (ANN and FFNN) and machine learning algorithms to classify them.

## Figures and Tables

**Figure 1 sensors-22-01629-f001:**
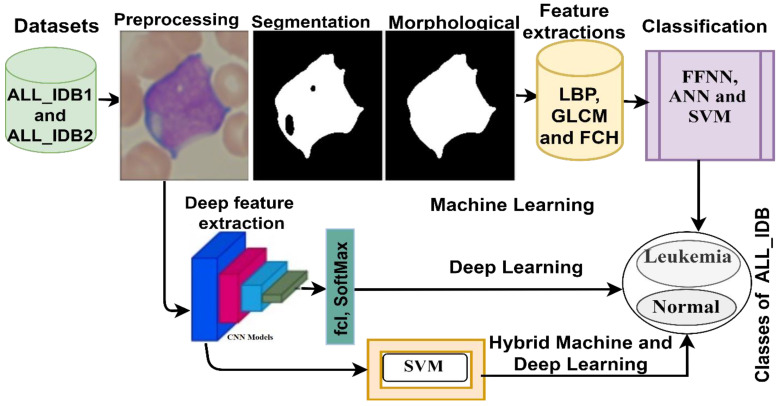
Morphological diagram for leukemia diagnosis from the ALL_IDB1 and ALL_IDB2 dataset.

**Figure 2 sensors-22-01629-f002:**
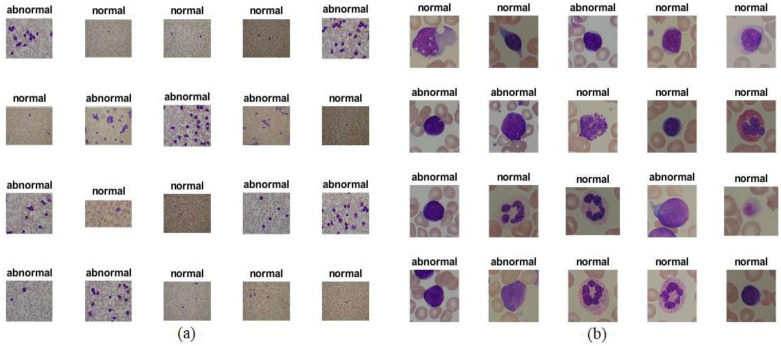
Samples from the (**a**) ALL-IDB1 and (**b**) ALL-IDB2 datasets.

**Figure 3 sensors-22-01629-f003:**
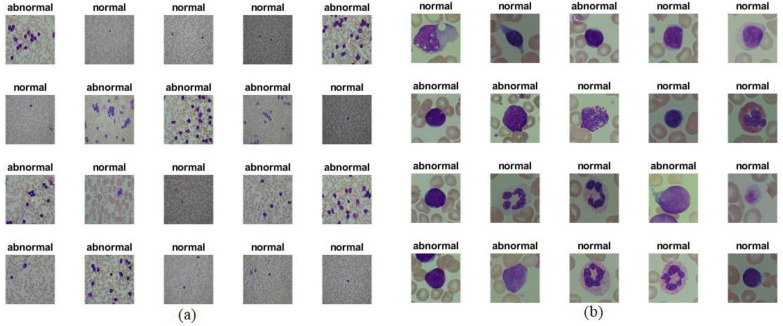
Samples from the (**a**) ALL-IDB1 and (**b**) ALL-IDB2 datasets after enhancement.

**Figure 4 sensors-22-01629-f004:**
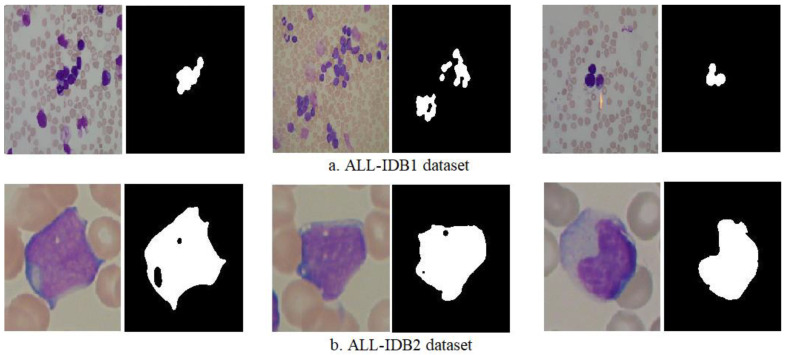
Samples from the two datasets after segmentation.

**Figure 5 sensors-22-01629-f005:**
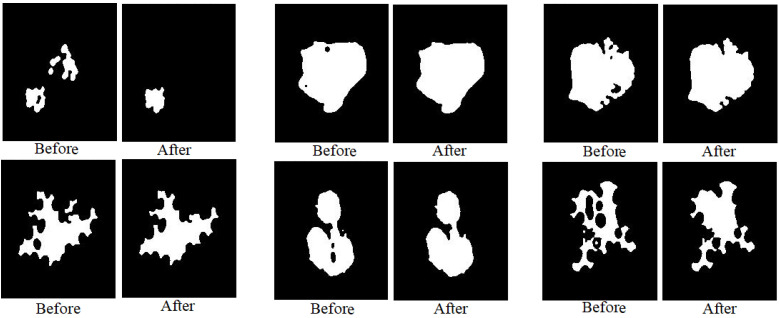
Samples from the two datasets before and after the morphological processes.

**Figure 6 sensors-22-01629-f006:**
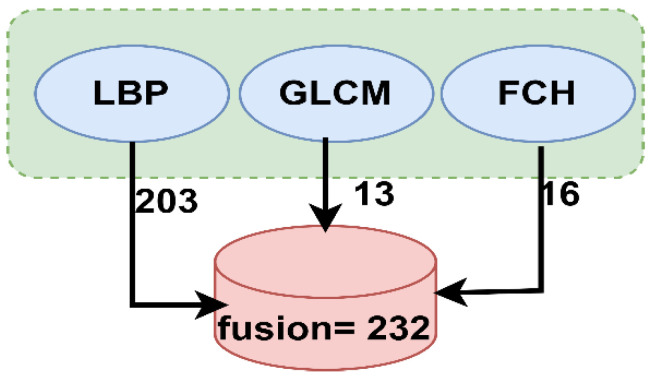
Fusion of features extracted by the LBP, GLCM, and FCH algorithms.

**Figure 7 sensors-22-01629-f007:**
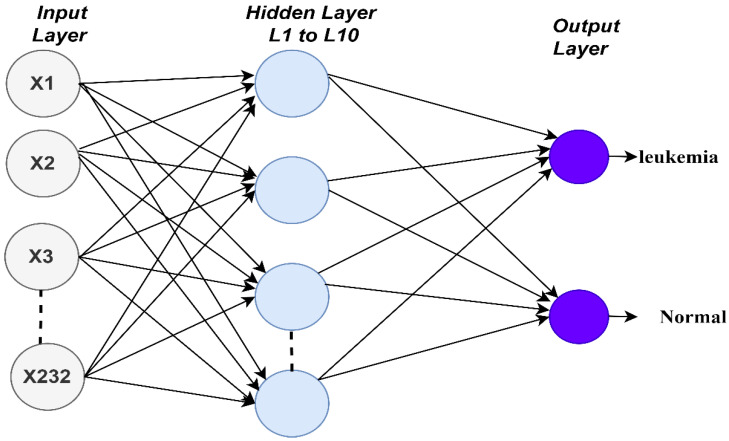
Architecture of the artificial neural network and feed forward neural network classifiers of the ALL-IDB dataset.

**Figure 8 sensors-22-01629-f008:**
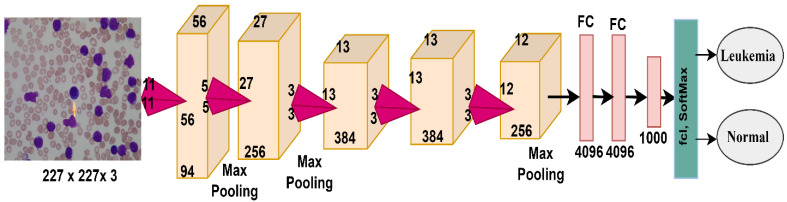
AlexNet network structure for ALL-IDB dataset diagnostics.

**Figure 9 sensors-22-01629-f009:**
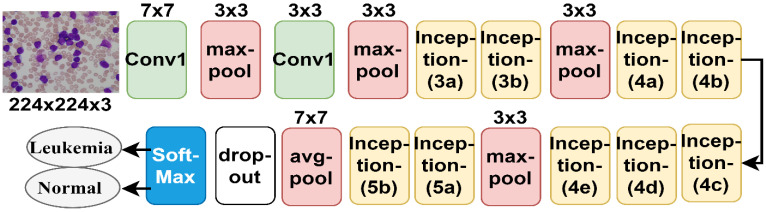
GoogLeNet network structure for ALL-IDB dataset diagnostics.

**Figure 10 sensors-22-01629-f010:**
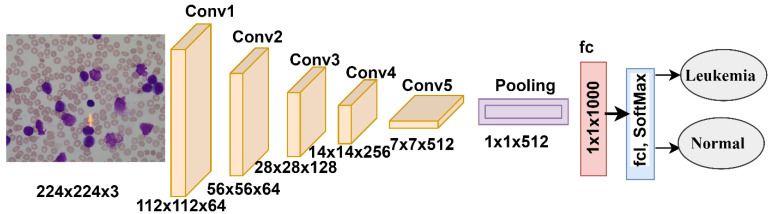
ResNet-18 network structure for ALL-IDB dataset diagnostics.

**Figure 11 sensors-22-01629-f011:**
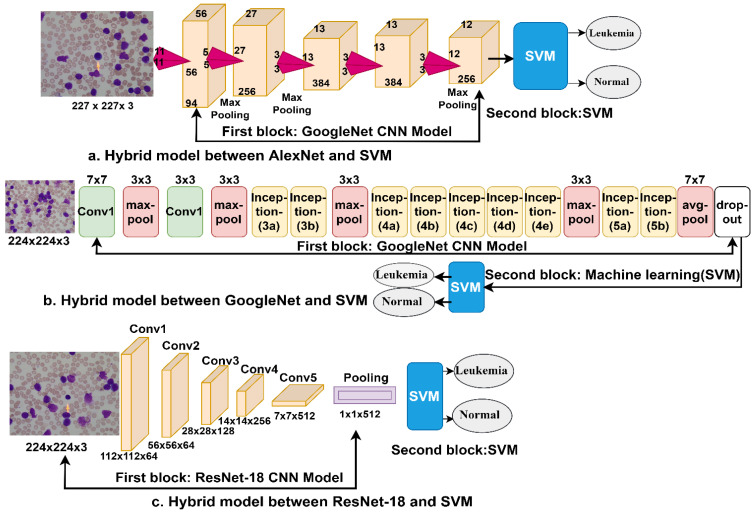
Deep learning–machine learning hybrid techniques: (**a**) AlexNet + SVM; (**b**) GoogleNet + SVM; and (**c**) ResNet-18 + SVM.

**Figure 12 sensors-22-01629-f012:**
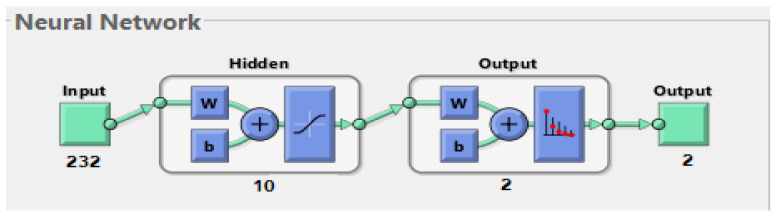
Training of the feed forward neural network classifier of the ALL_IDB1 and ALL_IDB2 datasets.

**Figure 13 sensors-22-01629-f013:**
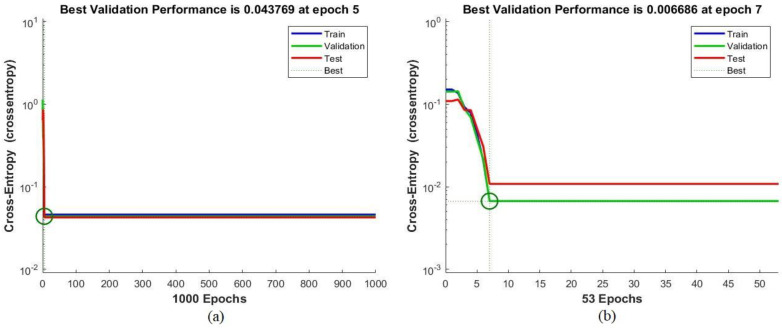
Performance of the artificial neural network classifier of the (**a**) ALL_IDB1 and (**b**) ALL_IDB2 datasets.

**Figure 14 sensors-22-01629-f014:**
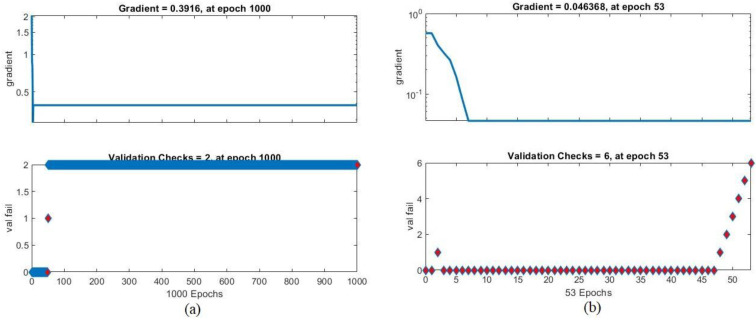
Gradient and validation check values of the (**a**) ALL_IDB1 and (**b**) ALL_IDB2 datasets.

**Figure 15 sensors-22-01629-f015:**
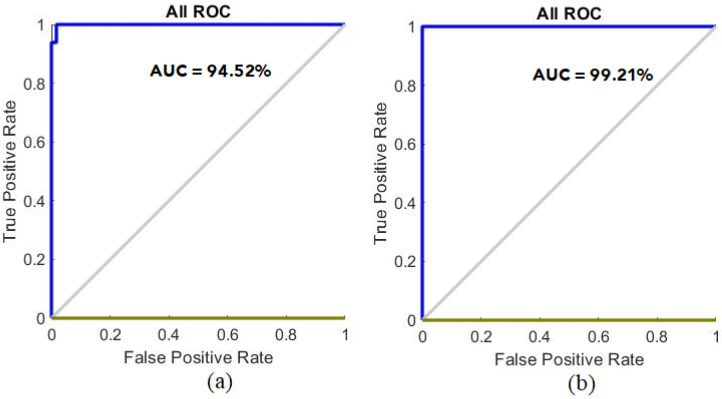
Receiver operating characteristic values of the (**a**) ALL_IDB1 and (**b**) ALL_IDB2 datasets.

**Figure 16 sensors-22-01629-f016:**
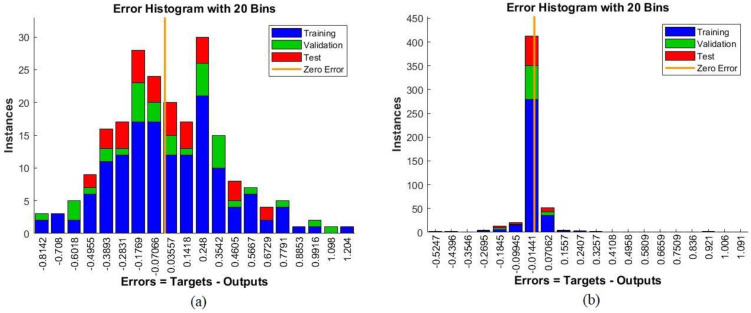
Error histograms of the (**a**) ALL_IDB1 and (**b**) ALL_IDB2 datasets.

**Figure 17 sensors-22-01629-f017:**
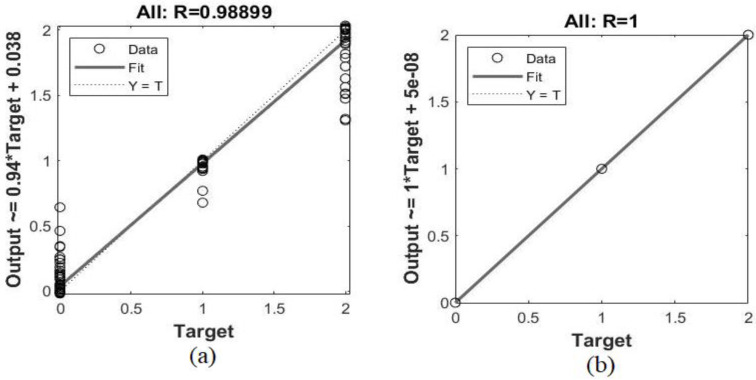
Regression of the (**a**) ALL_IDB1 and (**b**) ALL_IDB2 datasets.

**Figure 18 sensors-22-01629-f018:**
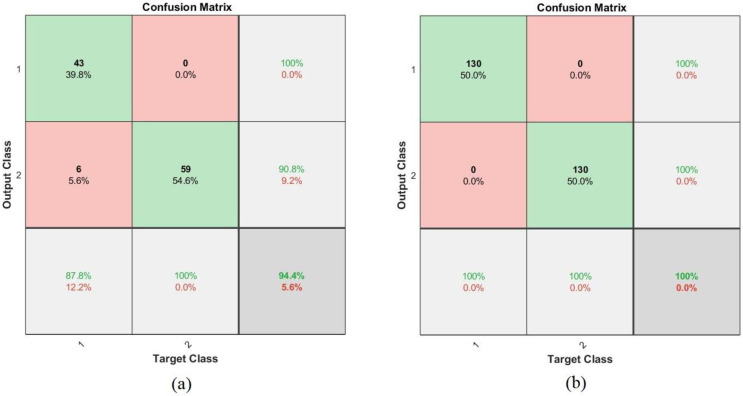
Confusion matrix for the ANN algorithm of the (**a**) ALL_IDB1 and (**b**) ALL_IDB2 datasets.

**Figure 19 sensors-22-01629-f019:**
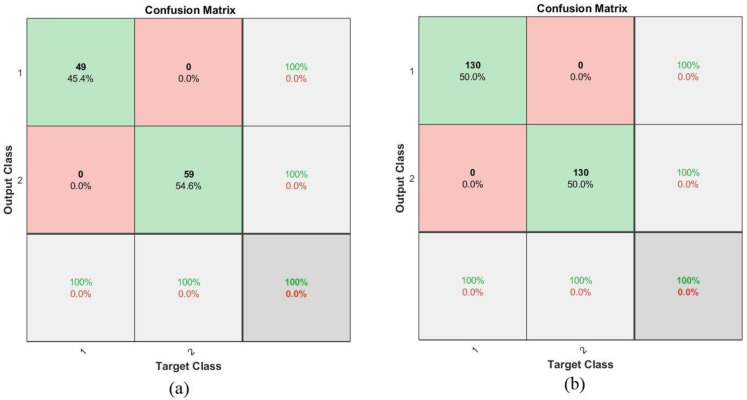
Confusion matrix for the FFNN algorithm of the (**a**) ALL_IDB1 and (**b**) ALL_IDB2 datasets.

**Figure 20 sensors-22-01629-f020:**
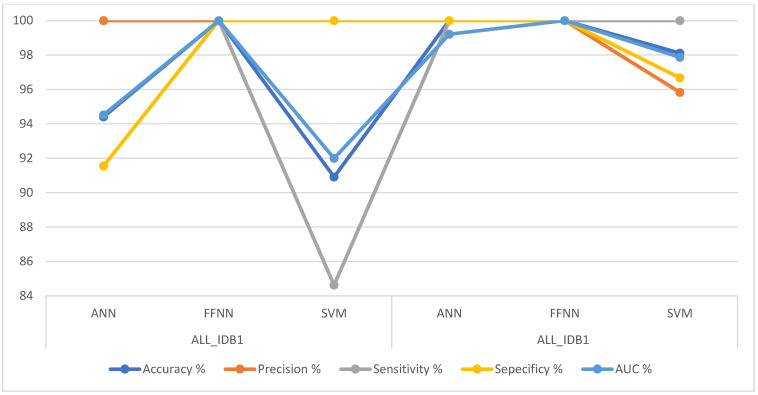
Performances of all the ANN, FFNN and SVM algorithms for the ALL_IDB1 and ALL_IDB2 datasets.

**Figure 21 sensors-22-01629-f021:**
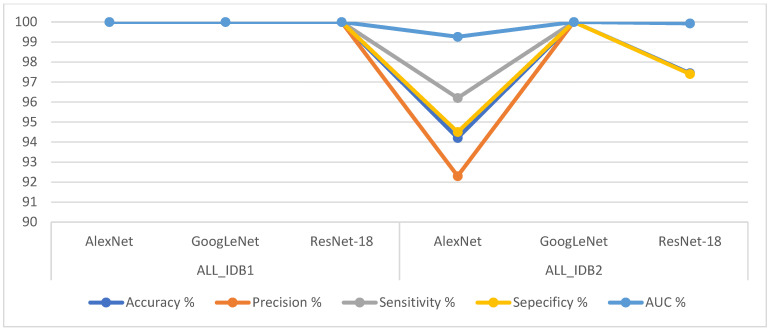
Performances of all the convolutional neural network models for the ALL_IDB1 and ALL_IDB2 datasets.

**Figure 22 sensors-22-01629-f022:**
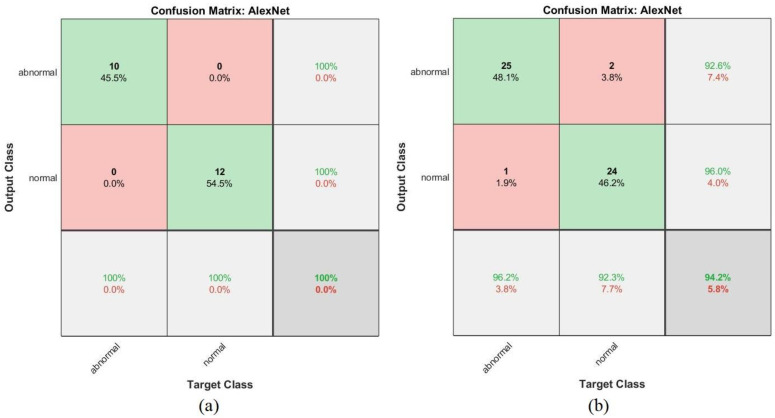
Confusion matrix for the AlexNet model for the (**a**) ALL_IDB1 and (**b**) ALL_IDB2 datasets.

**Figure 23 sensors-22-01629-f023:**
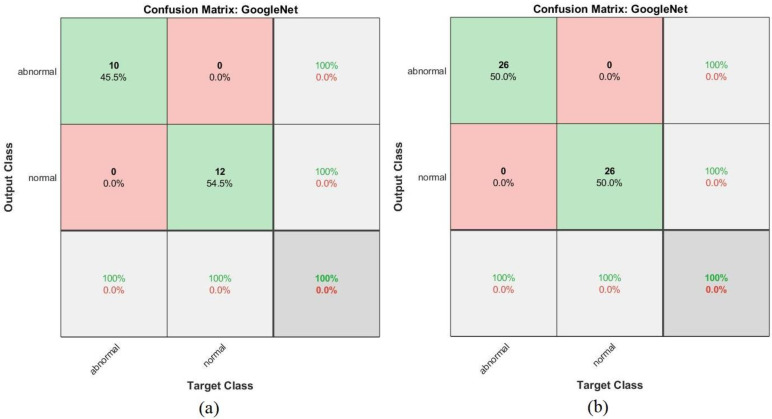
Confusion matrix for the GoogLeNet model for the (**a**) ALL_IDB1 and (**b**) ALL_IDB2 datasets.

**Figure 24 sensors-22-01629-f024:**
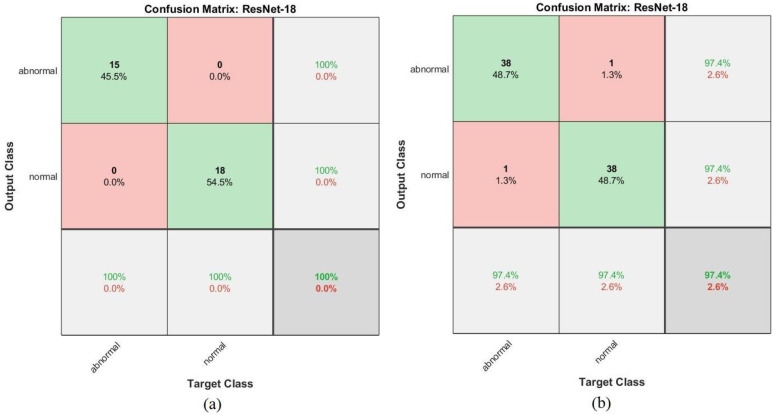
Confusion matrix for the ResNet-18 model for the (**a**) ALL_IDB1 and (**b**) ALL_IDB2 datasets.

**Figure 25 sensors-22-01629-f025:**
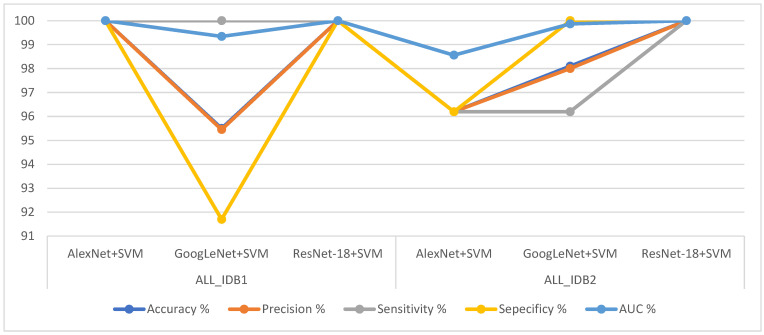
Performances of all the hybrid techniques for the ALL_IDB1 and ALL_IDB2 datasets.

**Figure 26 sensors-22-01629-f026:**
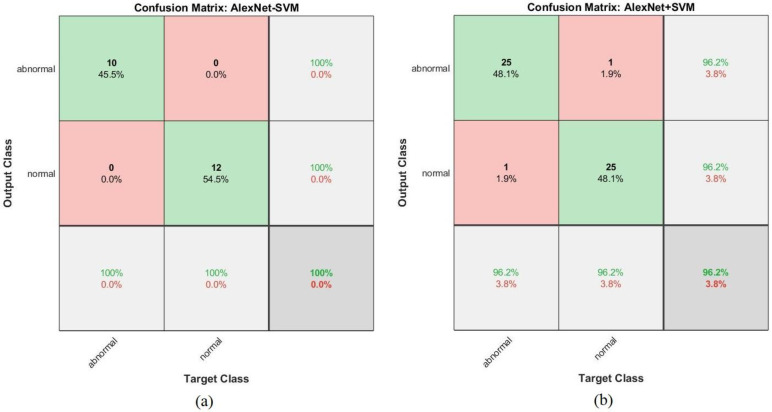
Confusion matrix for the AlexNet + SVM hybrid technique for the (**a**) ALL_IDB1 and (**b**) ALL_IDB2 datasets.

**Figure 27 sensors-22-01629-f027:**
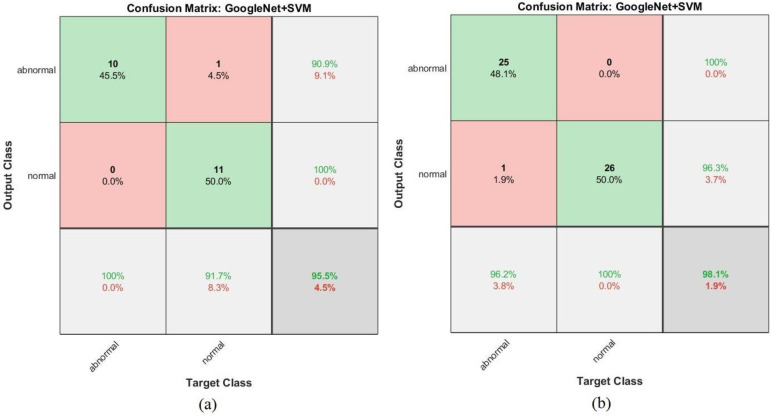
Confusion matrix for the GoogLeNet + SVM hybrid technique for the (**a**) ALL_IDB1 and (**b**) ALL_IDB2 datasets.

**Figure 28 sensors-22-01629-f028:**
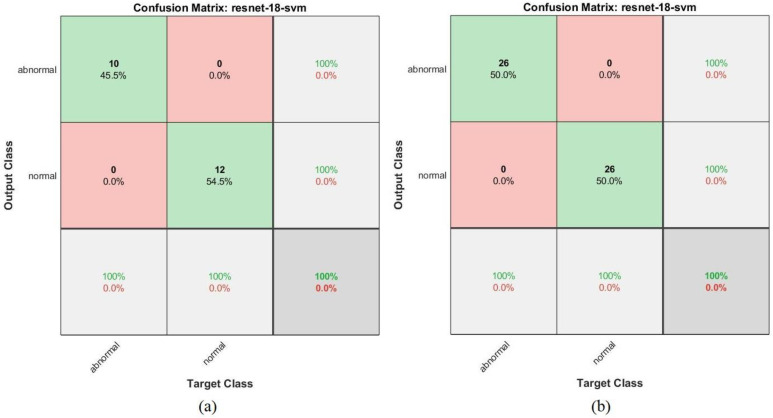
Confusion matrix for the ResNet-18 + SVM hybrid technique for the (**a**) ALL_IDB1 and (**b**) ALL_IDB2 datasets.

**Figure 29 sensors-22-01629-f029:**
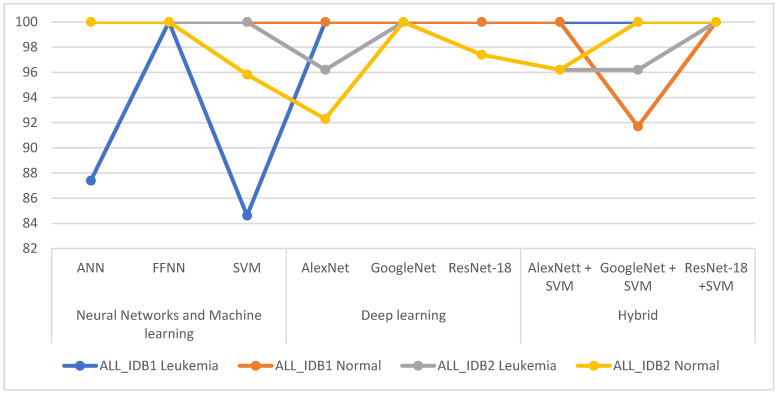
Comparison of the performances of the proposed systems for diagnosing leukemia from the two datasets.

**Figure 30 sensors-22-01629-f030:**
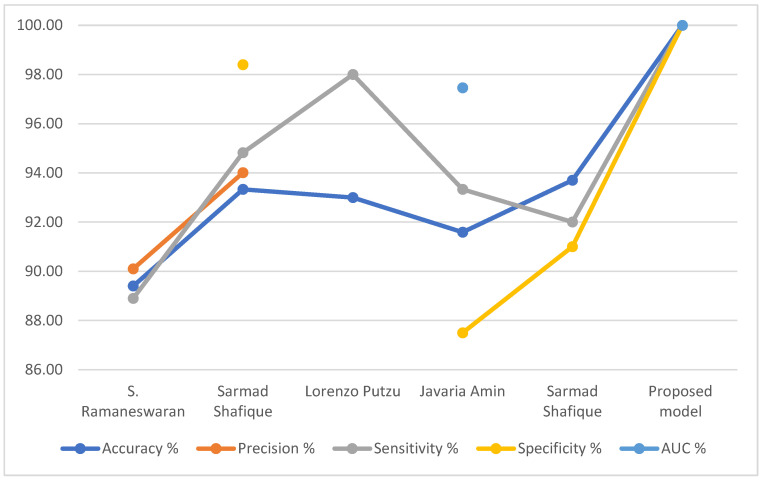
Comparing the performance of our proposed system with the previous systems.

**Table 1 sensors-22-01629-t001:** Splitting the ALL_IDB1 and ALL_IDB2 datasets for training and testing.

Dataset	ALL_IDB1			ALL_IDB2		
Phase	80% for training and validation (80:20%)		20% for testing	80% for training and validation (80:20%)		20% for testing
Classes	Training (80%)	validation (20%)		Training (80%)	validation (20%)	
Leukemia	31	8	10	83	21	26
Normal	38	9	12	83	21	26

**Table 2 sensors-22-01629-t002:** The performance of the ANN and FFNN algorithms on the ALL_IDB1 and ALL_IDB2 datasets.

Dataset	ALL_IDB1			ALL_IDB1		
Classifiers	ANN	FFNN	SVM	ANN	FFNN	SVM
Accuracy %	94.4	100	90.91	100	100	98.11
Precision %	100	100	100	100	100	95.83
Sensitivity %	91.55	100	84.62	100	100	100
Specificity %	91.55	100	100	100	100	96.67
AUC %	94.52	100	91.99	99.21	100	97.86

**Table 3 sensors-22-01629-t003:** Balancing ALL_IDB1 and ALL_IDB2 datasets during the training phase.

Dataset	ALL_IDB1		ALL_IDB2	
Phase	During the training phase		During the training phase	
Classes	Leukemia	Normal	Leukemia	Normal
No images before augmentation	39	47	104	104
No images after augmentation	**780**	**799**	**1040**	**1040**

**Table 4 sensors-22-01629-t004:** Adjust training parameters options for AlexNet, GoogLeNet and ResNet-18 models.

Options	AlexNet	GoogleNet	ResNet-18
training Options	adam	adam	adam
Mini Batch Size	130	20	15
Max Epochs	10	5	8
Initial Learn Rate	0.0001	0.0003	0.0001
Validation Frequency	50	3	5
Execution Environment	4 GB GPU	4 GB GPU	4 GB GPU

**Table 5 sensors-22-01629-t005:** The results of the AlexNet, GoogLeNet and ResNet-18 models on the ALL_IDB1 and ALL_IDB2 datasets.

Dataset	ALL_IDB1			ALL_IDB2		
Measure	AlexNet	GoogLeNet	ResNet-18	AlexNet	GoogLeNet	ResNet-18
Accuracy %	100	100	100	94.2	100	97.44
Precision %	100	100	100	92.3	100	97.4
Sensitivity %	100	100	100	96.2	100	97.4
Sepecificy %	100	100	100	94.5	100	97.4
AUC %	100	100	100	99.26	100	99.93

**Table 6 sensors-22-01629-t006:** The results of the AlexNet, GoogLeNet and ResNet-18 models on the ALL_IDB1 and ALL_IDB2 datasets.

Dataset	ALL_IDB1			ALL_IDB2		
Measure	AlexNet + SVM	GoogLeNet + SVM	ResNet-18 + SVM	AlexNet + SVM	GoogLeNet + SVM	ResNet-18 + SVM
Accuracy %	100	95.5	100	96.2	98.1	100
Precision %	100	95.45	100	96.2	98	100
Sensitivity %	100	100	100	96.2	96.2	100
Specificity %	100	91.7	100	96.2	100	100
AUC %	100	99.34	100	98.56	99.87	100

**Table 7 sensors-22-01629-t007:** The accuracy reached by proposed system in the diagnosis each class.

Dataset	Diseases	Neural Networks and Machine Learning			Deep Learning			Hybrid		
		ANN	FFNN	SVM	Alex-Net	Google-Net	Res-Net-18	AlexNett + SVM	GoogleNet + SVM	ResNet-18 + SVM
ALL_IDB1	Leukemia	87.4	100	84.62	100	100	100	100	100	100
	Normal	100	100	100	100	100	100	100	91.7	100
ALL_IDB2	Leukemia	100	100	100	96.2	100	97.4	96.2	96.2	100
	Normal	100	100	95.83	92.3	100	97.4	96.2	100	100

**Table 8 sensors-22-01629-t008:** Comparing the performance of the proposed systems with the performance of the relevant previous studies.

Previous Studies	Accuracy %	Precision %	Sensitivity %	Specificity %	AUC %
S. Ramaneswaran.; et al. [42]	89.40	90.10	88.90	-	83.20
Sarmad Shafique.; et al. [43]	93.33	94.01	94.82	98.4	-
Lorenzo Putzu.; et al. [44]	93.00	-	98.00	-	-
Javaria Amin.; et al. [45]	91.59	-	93.33	87.50	97.46
Sarmad Shafique.; et al. [46]	93.70	-	92.00	91	-
**Proposed model**	**100.00**	**100.00**	**100.00**	**100.00**	**100.00**

## Data Availability

In this study, the data of the two datasets ALL_IDB1 and ALL_IDB2 used to support the results of this study were collected from this link: https://www.kaggle.com/nikhilsharma00/leukemia-dataset (accessed on October 2021).

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
