# Peer review of "Multi-Method Diagnosis of Blood Microscopic Sample for Early Detection of Acute Lymphoblastic Leukemia Based on Deep Learning and Hybrid Techniques"

_sensors, 2022, doi:10.3390/s22041629_

Round 1

Reviewer 1 Report

Title: Multi-Method diagnosis of Blood Microscopic sample for Early Detection of Acute Lymphoblastic Leukemia Based on Deep Learning and Hybrid Techniques

Authors: Ibrahim Abunadi, and Ebrahim Mohammed Senan

Recommendation: Publish in present form.

Comments:

This is a solid work validating the application of AI algorithms for the early detection of leukemia. Three systems were proposed based on machine learning, deep learning, and hybrid techniques. The two Acute Lymphoblastic Leukemia Image Database (ALL_IDB1 and ALL_IDB2) were analyzed. This is relevant to the development of auto bioimaging analysis, and this work should generate broad interest.

My primary interests in reading the manuscript are with aspects of what key information and features the AI algorithms will be looking for to distinguish the lymphomas versus normal cells. The answer seems to be by the increased WBC volume/size and/or decreased RBC volume and platelet count. By naked eyes, all the images of normal vs. abnormal cells from the database 1 make sense (they are very visually different). However, the normal vs. abnormal cells from database 2 seems to have no prominent difference in their cell outline and volume (i.e., Figure 2b). Thus, would the input from Database 2 affect the overall accuracy of the proposed systems?

Several minor comments are listed below.

  1. I would rephrase and streamline the first paragraph of the Introduction section, as it provides too many detailed and general information about the WBC and RBC. I suggest the author to only focus on the features of normal cell vs. cells from the acute lymphoblastic leukemia patient.

  1. - In Figure 2, I suggest that the authors regroup the normal and abnormal subsets for both dataset 1&2.

- Again, the images from dataset 1&2 have very different appearance. The author should provide the reason/description why there is no consistency in cell shape/volume/size in dataset 1 vs. 2.

- All scale bars are missing.

  1. The information from Figure 3 is overwhelming and looks not prominently different from Figure 2. The best presentation will be picking one or two images and comparing their before/after enhancements, with arrows clearly indicating where the image spots has been enhanced.

  1. - In Figure 4, the segmentation seems to pick only partial of the cells stained in purple rather than all, can this be improved?

- How do the authors solve the issues of partially overlapping cells? It can be imaged that a wrong segmentation process could give false positive results because a large cell volume could also be resulted from several small cells adding up, thus yielding a large outline. I might think this literature could be helpful: "Segmentation, Inference and Classification of Partially Overlapping Nanoparticles".

  1. On Page 7, Line 9, please correct the sentence to “The processes test if the structure element “fits in” with its neighbors or not, and the “hits” process tests the intersection between neighbors.”.

  1. In Figure 20, I cannot see the overlapped precision line, maybe make this curve ub somewhat transparent.

  1. On Page 25, I believer the Table# is 6.

  1. On Page 27, last paragraph, I believe that the authors are referring to Table 7.

Reviewer 2 Report

In this work, authors developed diagnostic systems for the two Acute Lymphoblastic Leukemia Image Database (ALL_IDB1 and ALL_IDB2) for the early detection of leukemia. The following review comments are recommended, and the authors are invited to explain and modify.

Comment: The abstract section is inconsistent and does not reflect the main contributions of the manuscript. The authors should rewrite the abstract section to mention the main purpose of the paper, primary contributions, experimental results, and global implications.

Comment: What’s logic behind to present three proposed systems, why does thus the FC net used to be better? It is elegantly solved by CNN, which can effectively process.

Comment: When writing phrases like “As medical images contain thousands of data that are difficult to analyze with high efficiency, feature extraction is one of the essential stages,” it should cite some related work in order to sustain the statement (perhaps, just adding some manuscripts cited in the first part of the paper).

Comment: Novelty is confusing. A highlight is required. The main contributions of the manuscript are not clear. The main contributions of the ‎article must be very clear and would be better if summarize ‎them into 3-4 points at the ‎end of the introduction.‎

Comment: Authors should be clearer on why they set all of the parameters shown in Table 4 in that way. I understand you reached that values after many experiments, but you should be more verbose on why you chose that final values shown in Table 4.

Comment: Results are not clearly compared with the state-of-art. This point is fundamental in scientific research and it is the main reason for a major revision.

Comment: Mention the limitations and future works of the developed system elaborately.

Comment: Moreover, it should be noticed that the clinical appliance has to be decided by medicals since the existing differences between the real image and the one generated by the proposed system could be substantial in the medical field.

Comment: Could you please check your references carefully (in particular, proceedings: location of the conference, date of the conference, publisher's name and location...)? All references must be complete before the acceptance of a manuscript.

Round 2

Reviewer 2 Report

The authors have answered my questions satisfactorily.